# Study on Dynamics of Overrunning Spring Clutches and Suppression Methods for Their Abnormal Noise

Jie Zhou [1,†], Zhehang Qiu [1,2,†], Huijuan Zhang [1] and Jianming Zhan [1,*]

1    School of Mechatronics and Energy Engineering, NingboTech University, Ningbo 315100, China;
     zhoujie@nbt.edu.cn (J.Z.); 22225135@zju.edu.cn (Z.Q.); zhanghuijuan@nit.net.can (H.Z.)
2    School of Mechanical Engineering, Zhejiang University, Hangzhou 310058, China
*    Correspondence: zhanjm@nit.zju.edu.cn
†    These authors contributed equally to this work.

**Abstract:** Overrunning spring clutches are widely used as essential transmission devices, and the occurrence of abnormal noise can lead to a decline in their performance. This study investigates the dynamic aspects of abnormal noise in engineering applications, including its causes, influencing factors, and suppression methods. Audio processing algorithms are employed to analyze the audio associated with abnormal noise, and the Fourier Motion Blur algorithm is applied to process video images of the springs. By combining the motion blur curve with the noise spectrum curve, the source of the abnormal noise is identified as friction-induced vibrations in the spring. Theoretical modeling and calculations are carried out from a dynamic perspective to validate that the phenomenon of abnormal noise in the clutch is a result of self-excited friction vibration caused by the stick–slip phenomenon. Based on theoretical analysis and practical engineering, surface texturing is added to the center shaft of the spring seat, optimizing the system as an overdamped system to suppress self-vibration. Utilizing CFD simulation analysis, the simulation results are used to improve the texturing parameters and further optimize the texturing shape, resulting in an optimal parallelogram surface texture structure. Experimental validation confirms that the improved overrunning spring clutch completely eliminates abnormal noise during overrunning operation. Therefore, this paper contributes to the understanding of the dynamic issues associated with abnormal noise in overrunning spring clutches, confirming that the mechanism for abnormal noise generation is friction-induced self-excitation vibration, and demonstrating that surface texture optimization methods effectively suppress the occurrence of abnormal noise.

**Keywords:** overrunning spring clutches; abnormal noise; friction-induced self-excited vibration; dynamic analysis

## 1. Introduction

Overrunning spring clutches are components that transmit torque and control engagement through the deformation of a torsion spring. When torque is being transmitted, the spring contracts, causing strong frictional torque that automatically engages the primary and secondary shafts of the clutch. In the case of reverse overrunning, the frictional torque is reduced, leading to the automatic disengagement of the primary and secondary shafts [1]. However, in current engineering applications, the reverse overrunning process of overrunning spring clutches often gives rise to significant abnormal noise issues. Abnormal noise directly impacts the performance of these clutches, making it crucial to study, analyze, and optimize these noise-related problems in the clutch.

Considerable research has examined the torsion spring structure in overrunning spring clutches. Pan et al. [2] conducted a preliminary study on the working principle and mechanical behavior of an overrunning spring clutch from a mechanical perspective. Based on the force analysis and stress analysis of the overrunning spring clutch, they

proposed effective calculation methods, providing a reference for the stress analysis of the spring. Further theoretical optimization, led by researchers like Chang [3], involved critical spring parameter calculations and comprehensive optimization techniques, forming the foundation for clutch model development. However, the aforementioned studies primarily focus on the static stress analysis of the clutch's spring, which has certain limitations. Moreover, the emphasis during model development has largely been on optimizing key spring parameters, with insufficient attention given to the comprehensive analysis of the entire component system.

Friction is a constant factor in the operation of overrunning clutches, yet the research on how friction impacts the dynamic performance of overrunning spring clutches is relatively limited. Yan [4] established a numerical model for a variable cross-section torsion spring clutch. Their primary focus was on studying the impact of key structural parameters, such as the spring clutch's variable cross-section spring clearance and the thickness of its excitation ring, on the torque transmission performance. Subsequently, they analyzed the stress within and outside the spring under different structural parameters as well as the failure characteristics of spring clutches in torque transmission. They also conducted an analysis of the frictional wear characteristics in overrunning spring clutches, concluding that frictional wear is a persistent issue during reverse overrunning. Using the Archard wear calculation model and a combined wear calculation method, they calculated the wear life of overrunning spring clutches [5]. However, it is important to note that these studies predominantly address the service life of clutches in relation to frictional wear. Further analysis is warranted to comprehensively comprehend the vibrations and noise arising from the continuous friction occurring during clutch operation in applications.

The operational state of overrunning spring clutches has a direct impact on the overall system's stability, underscoring the pivotal need for dynamic analysis. Researchers such as Yan and Long [6] analyzed the working principles of overrunning spring clutches. They established finite element models for these clutches and employed finite element analysis to investigate the dynamic characteristics of variable cross-section overrunning spring clutches. Cao et al. [7,8] conducted research on the engagement dynamic characteristics of an overrunning spring clutch, establishing a dynamic model of the clutch engagement state. They discussed the physical parameters determining the torque during clutch engagement and conducted dynamic experiments on clutch engagement. However, their work primarily focused on the dynamics of clutch engagement and did not provide substantial reports on the motion characteristics during reverse overrunning.

Most of the existing research has predominantly focused on mechanical modeling and friction analysis of overrunning spring clutches, optimizing critical parameters of the torsion spring component. However, there is a noticeable absence of research describing optimization methods for addressing the dynamic abnormal noise issues in overrunning spring clutches. In the field of manufacturing and related domains, surface micro-texturing methods [9–11] are commonly applied for friction optimization. Liu et al. [12] used laser technology to create three different shapes of laser micro-textures on the surface of hot-rolled steel rail rollers to study the friction performance of wheel–rail steel. They suggested that the injection of lubricating grease plays a predominant role in reducing the coefficient of friction, and the lubrication effect is closely related to the shape of the laser micro-texture, with rectangular and rhombic textures exhibiting superior lubrication. Seid Ahmed et al. [13] utilized a titanium sapphire laser to generate various geometric textures on austenitic stainless steel, and through experimental investigation, it was concluded that the surfaces treated with textures exhibited lower friction coefficients and better wear resistance compared to untreated surfaces. However, in practical engineering applications, the bases of overrunning spring clutches are often injection-molded components, and laser-based surface micro-texturing methods present certain issues, including thermal effects [14,15], the formation of debris [16], and the creation of dimples [17]. Additionally, in the production of overrunning spring clutches, the processing method of laser texturing involves the secondary processing of injection-molded parts, thereby causing the processing efficiency

of laser texturing to be lower than that of one-time injection molding [18]. Similarly, mechanical micro-texturing methods have limitations in terms of size and precision. Therefore, surface texturing methods can be utilized for the optimization of dynamic abnormal noise in overrunning spring clutches, with a focus on configuring parameters that are conducive to one-piece injection molding for practical engineering applications.

This paper investigates the dynamic principles and manifestations related to the causes, influencing factors, and suppression methods of abnormal noise in overrunning spring clutches during engineering applications. Experimental measurements confirm that the source of abnormal noise stems from self-excited friction vibration caused by the stick–slip phenomenon. A comprehensive dynamic model was established to analyze the mechanism behind the vibration. During the clutch's reverse overrunning state, the system's driving speed falls below the critical speed threshold for friction-induced self-excitation vibrations, leading to the occurrence of abnormal noise. Subsequently, surface texture optimization was applied to the connecting seat of the overrunning spring clutch based on the characteristics of the system's dynamic model. Meeting the requirements of the one-piece injection molding process, CFD simulations were used to determine that a parallelogram with a width of 0.95 mm is the optimal texture structure parameter. Several sets of overrunning spring clutch samples were subjected to bench testing before and after optimization, validating the effective suppression of abnormal noise due to surface texture optimization.

## 2. Analysis of Abnormal Noise Causes

This study investigates the occurrence of abnormal noise in overrunning spring clutches when used in lifting systems. Figure 1 illustrates the structural components and operational principles of the clutch system. The clutch is composed of a torsion spring, a connecting seat, and a fixed seat. The connecting seat consists of a central shaft and a bottom base, both of which are secured to the motor. One end of the torsion spring is rigidly connected to the fixed seat, and the coil is meshed with the central shaft through an interference fit. Contact between the torsion spring and the base provides positioning. During the overrunning state, the torsion spring radially expands, reducing the frictional torque to achieve overrunning. In the engaged state, the torsion spring radially tightens, increasing the frictional torque, thereby enabling self-locking protection. In practical applications, overrunning spring clutches may experience abnormal noise during the upward stroke, resulting in reduced efficiency and potential damage to the clutch.

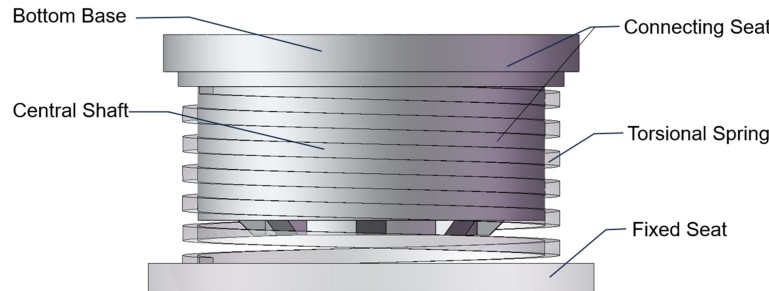

**Figure 1.** The structure of overrunning spring clutch.

### 2.1. Phenomenon Analysis

The occurrence of abnormal noise in overrunning spring clutches within lifting systems manifests as periodic sounds at low speeds. Typical abnormal sound frequencies were obtained through a sound sensor, and MATLAB R2022b was utilized for audio visualization and processing. The sampling frequency was set to 8000 Hz, and the time-domain waveform of the abnormal noise is depicted in Figure 2.

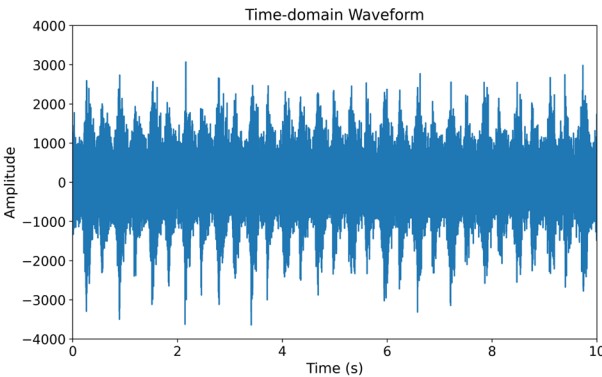

**Figure 2.** The waveform of abnormal noise.

Due to significant speed fluctuations in the audio, which pose challenges in measurement using motion sensitivity, a high-speed camera with a frame rate of 360 Hz was employed to capture images of the overrunning spring clutches within the lifting system. Specific sections of the torsion spring components were marked with black notations. Sequential frames were selected, capturing instances when abnormal noise occurred. Some of these frames showed anomalies in the torsion spring section. Figure 3a represents a normal image, while Figure 3b shows an image with ghosting anomalies. The ghosting in the marked section resulted from the spring's vibration, a phenomenon that can be described using motion blur, arising from the relative movement between the camera and the subject being photographed [19].

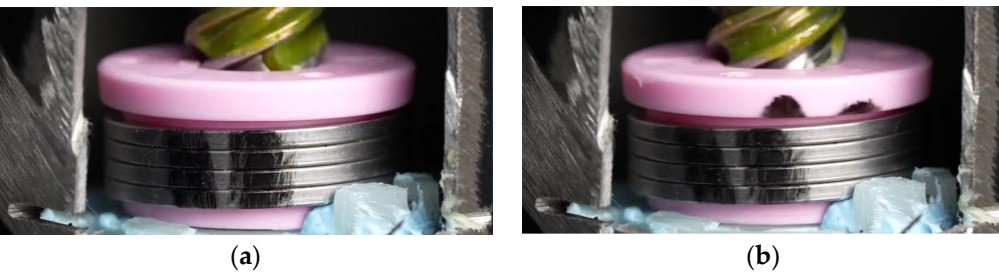

| (**a**) | (**b**) |

**Figure 3.** High-speed photography comparison of normal and abnormal clutch behavior. Normal clutch behavior (**a**). Abnormal clutch behavior (**b**).

Motion blur was calculated from the high-speed video using image processing techniques. The video frames were converted into grayscale images through image processing algorithms, with $I(x, y)$ representing the grayscale pixel intensity of the image at pixel coordinates $(x, y)$. Subsequently, a two-dimensional Fourier transform was applied to the grayscale image, converting the image from the spatial domain to the frequency domain, represented as $F(u, v)$ at coordinates $(u, v)$. The square of the amplitude of each frequency component in the spectrum was computed using Equation (1):

$$E(u,v) = |F(u,v)|^2 \tag{1}$$

$E(u, v)$ represents the energy at the $(u, v)$ coordinates in the frequency domain. A lower concentration of energy in the spectrum indicates higher blur. The calculation of phase information in the spectrum was utilized to determine the main frequency direction, as shown in Equation (2):

$$P(u,v) = \arctan\left(\frac{Im(F(u,v))}{Re(F(u,v))}\right) \tag{2}$$

$Im(F(u, v))$ represents the imaginary part at coordinates $(u, v)$ in the spectrum, while $Re(F(u, v))$ represents the real part. $P(u, v)$ denotes the phase at coordinates $(u, v)$ in the frequency domain. The Fourier Motion Blur metric is defined as the weighted average of the energy in the main frequency direction in the spectrum, as shown in Equation (3):

$$K = \frac{1}{N} \sum_{u,v} E(u,v) \cdot \cos^2(P(u,v) - \theta) \tag{3}$$

where $N$ is the size of the spectrum, and $\theta$ represents the main frequency direction. Figure 4a,b display the resulting blurry image and the corresponding audio image of the processed video, corresponding to the abnormal sound discussed in the previous section.

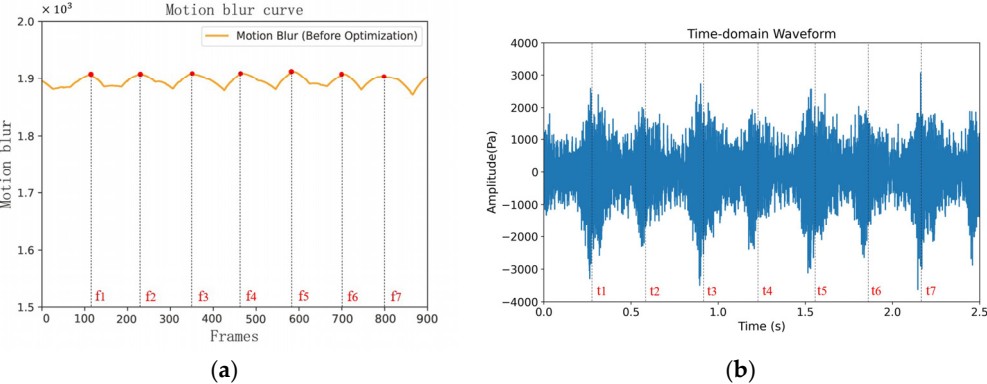

(**a**)                                                 (**b**)

**Figure 4.** Blur curve of torsion spring clutch in abnormal noise state. The motion blur of the abnormal overrunning spring clutches (**a**). The audio waveform of the abnormal overrunning spring clutches (**b**).

With a frame rate of 360 Hz, the peak times of the waveforms in Figure 4a,b are listed in Table 1. The peak times of the image correspond to those of the audio waveform, indicating a close relationship between the vibration of the torsion spring and the occurrence of abnormal noises. Notably, the peak times in the image align with those in the audio representation, suggesting periodic vibrations during the torsion spring's operation. Furthermore, several predominant anomalies are observed during abnormal noise in the clutch, including noticeable resistance when rotating the torsion spring in reverse and severe friction at the tail of the torsion spring, leading to wear in the connecting seat. An analysis of these phenomena revealed that the primary source of the clutch's abnormal noise lies in the vibration of the torsion spring component. The vibration results from force excitation, with only frictional forces and fixed-end interaction forces acting on the torsion spring component during its operation. While the fixed-end interaction forces remain relatively stable, the frictional forces during the torsion spring's operation are intense and have a significant impact. Therefore, the abnormal noise in the torsion spring component is attributed to the vibrations generated by the frictional forces.

**Table 1.** Comparison of peak times between blur waveform and audio waveform.

| | Blur Curve | | Time-Domain Waveform | |
|---|---|---|---|---|
| Peak Identification | Peak Frame Number | Time (s) | Peak Identification | Time (s) |
| f1 | 108 | 0.30 | t1 | 0.29 |
| f2 | 221 | 0.61 | t2 | 0.58 |
| f3 | 348 | 0.97 | t3 | 0.94 |
| f4 | 465 | 1.29 | t4 | 1.24 |
| f5 | 585 | 1.63 | t5 | 1.56 |
| f6 | 700 | 1.94 | t6 | 1.87 |
| f7 | 799 | 2.22 | t7 | 2.21 |

### 2.2. Mechanical Analysis

The stress analysis and modeling of the torsion spring clutch system are depicted in Figure 5, showcasing the stress distribution within the torsion spring.

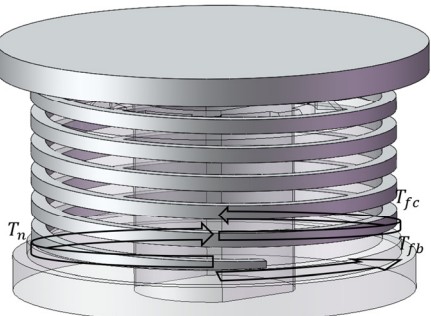

**Figure 5.** Overall force analysis diagram of the torsion spring clutch.

During the process of overrunning operation, the friction torque between the inner side of the torsion spring and the center shaft is denoted as $T_{fc}$, the friction torque between the bottom end of the torsion spring and the connecting seat is denoted as $T_{fb}$, while $T_n$ represents the elastic torque due to the deformation of the torsion spring itself. Each torque is calculated separately. To analyze the elastic torque $T_n$ resulting from torsion spring deformation, a model was established. As shown in Figure 6, based on the design characteristics of the torsion spring, the relationship between the torsion spring's elastic torque $T_n$ and the deformation angle $\varphi$ is described by Equations (4) and (5).

$$\varphi = \frac{Ml}{EI_0} = \frac{\pi M D_0 n}{EI_0} \approx \frac{180 T_n D_0 n}{EI_0} \tag{4}$$

$$T_n = \frac{EI_0 \varphi}{180 D_0 n} \tag{5}$$

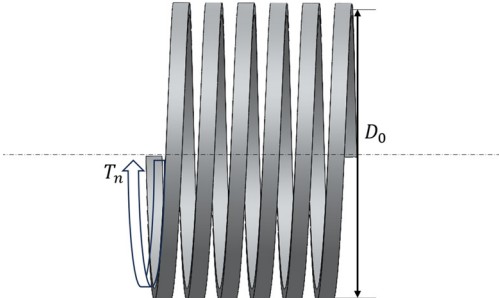

**Figure 6.** Analysis diagram of torsional spring elastic torque.

$M$ represents the torsional moment of the coil spring, $l$ denotes the length of the coil spring, $D_0$ stands for the mean diameter of the coil spring, $E$ represents the material elastic modulus, $I_0$ denotes the cross-sectional moment of inertia of the coil spring wire, and $n$ signifies the number of coil turns. It is evident that $T_n$ is directly proportional to $\varphi$.

Analyzing the frictional torque $T_{fc}$ between the torsion spring and the center shaft, considering the presence of lubricant, the friction between the torsion spring and the center shaft is a combination of both dry friction and viscous friction. $T_{fc}$ can be expressed using Equation (6):

$$T_{fc} = T_{fc1} + T_{fc2} \tag{6}$$

$T_{fc1}$ corresponds to the dry friction torque, while $T_{fc2}$ represents the damping friction torque. The calculation of $T_{fc2}$ is shown as follows in Equation (7):

$$T_{fc2} = -c(\dot{\varphi} - \omega_0) \tag{7}$$

$c$ represents the system damping, and $\varphi$ denotes the deformation velocity of the torsion spring.

To determine the magnitude of $T_{fc1}$, a simplified force analysis of a certain torsion spring coil is depicted in Figure 7. Within the cross-section of the torsion spring wire, there are tensile forces, bending moments, and negligible shear forces. Consider a small segment of the torsion spring wire, $dl$, corresponding to a central angle of $d\alpha$. The bending moments on the left and right sides are denoted as $M$, and the tensile forces are $F$ and $F + dF$, with $dF$ being a differential force. Additionally, the contraction force of the small torsion spring wire segment is $dP$, and the axial support force on the small torsion spring wire segment is $dN$. The friction coefficient between the center shaft and the torsion spring is denoted as $\mu_0$.

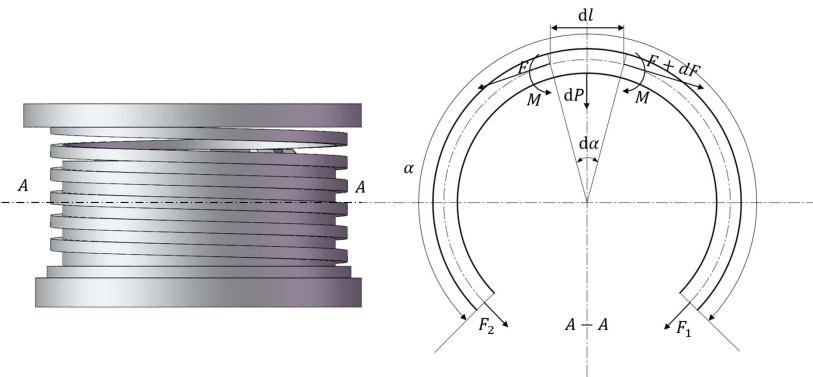

**Figure 7.** Force analysis diagram of a single torsional spring coil.

The equilibrium equations for the various forces on the small torsion spring wire segment in both the horizontal and vertical directions are as follows in Equations (8) and (9):

$$\mu_0 dN + (F + dF) \cos\frac{d\alpha}{2} - F \cos\frac{d\alpha}{2} = 0 \tag{8}$$

$$dP + F \sin\frac{d\alpha}{2} + (F + dF) \sin\frac{d\alpha}{2} - dN = 0 \tag{9}$$

The relationship between the bending moment $M$ and the contraction force $dP$ of the torsion spring wire is given by Equation (10):

$$dP = \frac{2Md\alpha}{D} \tag{10}$$

Since $d\alpha$ is very small, we can approximate it as follows: $\sin\frac{d\alpha}{2} = \frac{d\alpha}{2}$, $\cos\frac{d\alpha}{2} = 1$, $dF \sin\frac{d\alpha}{2} = 0$. Substituting Equations (10) and (8) into Equation (9), we obtain Equation (11):

$$\frac{dF}{\frac{2M}{D} + F} = -\mu_0 d\alpha \tag{11}$$

The torsion spring coil can be approximated as an open circular ring. Let the tension in any two cross-sections of the torsion spring coil be denoted as $F_1$ and $F_2$, and let $\alpha$ represent the angle between these two cross-sections. When $\alpha = 0$, $F_2 = 0$, and the effective tension

is given by $F = F_1 - F_2 = F_1$ [20]. Substituting this into Equation (8) and integrating, the friction torque for a single coil can be expressed as follows in Equation (12):

$$T_{fc1} = F\frac{D}{2} = M\left(e^{-f\alpha} - 1\right) \tag{12}$$

For the torsion spring clutch, $M = \frac{T_n}{e^{2\pi n f} - 1}$ [7], where $T_n$ represents the elastic torque, and for a spring coil with $n$ turns, the friction torque is given by Equation (13):

$$T_{fc1} = \frac{T_n\left(1 - e^{-2\pi\mu_0 n}\right)}{e^{2\pi\mu_0 n} - 1} = \frac{EI_0\varphi}{180D_0 n}\frac{1 - e^{-2\pi\mu_0 n}}{e^{2\pi\mu_0 n} - 1} \tag{13}$$

The relationship between the friction torque $T_{fc1}$ in the torsion spring clutch and the elastic torque $T_n$ is observed to be directly proportional, thus indicating a proportionality with respect to the torsion angle $\varphi$.

Regarding $T_{fb}$, the contact between the bottom of the spring and the connecting seat can be approximated as an open ring with a central angle $\beta_0$, which varies with motion. The pressure between the spring and the connecting seat is $F_N$. The large radius of the spring is $R_2$, and the small radius is $R_1$. According to the Stribeck friction model, dry friction occurs when the relative speed between the torsion spring and the base is 0, characterized by a static friction coefficient of $\mu_1$. When the relative speed is non-zero, it transitions to kinetic friction, where the friction coefficient $\mu_1'$ is less than $\mu_1$. The pressure per unit contact area is given by Equation (14):

$$P = \frac{4F_N}{\beta_0\pi\left(R_2{}^2 - R_1{}^2\right)} \tag{14}$$

Considering a ring element with a width of d$r$, as depicted in Figure 8, the frictional torque for the ring element during the clutch's initiation phase is expressed by Equation (15):

$$\mathrm{d}T_{fb} = \frac{r\beta_0}{2\pi}\frac{4F_N}{\beta_0\pi\left(R_2{}^2 - R_1{}^2\right)}2\pi r(\mu_1 - \mu_1')\mathrm{d}r = \frac{2\pi F_N(\mu_1 - \mu_1')}{\pi\left(R_2{}^2 - R_1{}^2\right)}r^2\mathrm{d}r \tag{15}$$

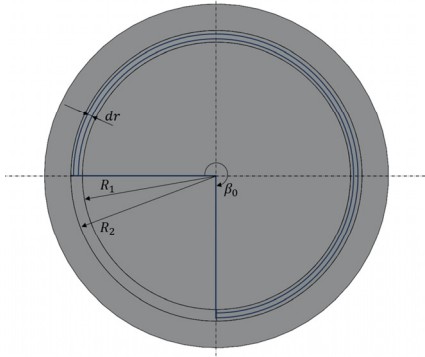

**Figure 8.** Force analysis diagram of the contact portion between the torsional spring and the base.

By integrating Equation (15), we can obtain $T_{fb}$, as shown in Equation (16):

$$T_{fb} = \int_{R_1}^{R_2} dT_{fb} = \frac{2(\mu_1 - \mu_1')F_N}{3\left(R_2{}^2 - R_1{}^2\right)}\left(R_2{}^3 - R_1{}^3\right) = \frac{2(\mu_1 - \mu_1')F_N\left(R_2{}^2 + R_1 R_2 + R_1{}^2\right)}{3(R_1 + R_2)} \tag{16}$$

It can be observed that the friction torque $T_{fb}$ is independent of the central angle $\beta_0$ of the open ring, and it can be considered as a constant torque.

*2.3. Dynamics Analysis*

Assuming the ground as the reference frame, with a positive direction of rotation for the torsion spring, a central axis rotation speed of $\omega_0$, and a rotational inertia of $I$, the relative displacement is given by $\varphi - \omega_0 t$. Based on the torque calculations above, the dynamic equation for the deformation of the torsion spring is given by Equation (17) [21]:

$$\boldsymbol{T}_{fb} sign(\dot{\varphi} - \omega_0) + \boldsymbol{T}_{fc} + \boldsymbol{T}_n + I\ddot{\varphi} = 0 \tag{17}$$

After substitution and rearrangement, the equation is as follows:

$$I\ddot{\varphi} + c\dot{\varphi} + k_1\varphi = \boldsymbol{T}_{fb} sign(\dot{\varphi} - \omega_0) + c\omega_0 + k_1\omega_0 t \tag{18}$$

The equivalent stiffness $k_1 = \left(1 + \frac{1 - e^{-2\pi\mu_0 n}}{e^{2\pi\mu_0 n} - 1}\right)\frac{EI_0}{180D_0 n}$, and $k_1$ increases as the friction coefficient $\mu_0$ decreases. This equation represents a second-order linear constant-coefficient nonhomogeneous differential equation, and its general solution is:

$$\varphi = \varphi_c + \varphi_p \tag{19}$$

$\varphi_c$ corresponds to the solution of the following homogeneous equation:

$$I\ddot{\varphi} + c\dot{\varphi} + k_1\varphi = 0 \tag{20}$$

The solution of Equation (20) is given by:

$$\varphi_c = e^{-\xi f_n t}(A \sin f_n t + B \cos f_n t) \tag{21}$$

where the damping ratio $\xi = \frac{c}{2\sqrt{k_1 I}}$, the natural frequency $f_n = \sqrt{\frac{k_1}{I}}$, and the particular solution is given by:

$$\varphi_p = \omega_0 t + \frac{\boldsymbol{T}_{fb} sign(\dot{\varphi} - \omega_0)}{k_1} \tag{22}$$

Considering $t = 0$, at which point the velocity $\dot{\varphi} = 0$ and the spring has not yet deformed, the acceleration can be approximated as $\ddot{\varphi} = \frac{T_{fb}}{I}$. In practical applications, the damping ratio $\zeta^2 \approx 0$, the solution is given by:

$$A = -\frac{\omega_0}{f_n}(F\xi + 1), B = \frac{\omega_0}{f_n}(2\xi - F) \tag{23}$$

$$F = \frac{\boldsymbol{T}_{fb} sign(\dot{\varphi} - \omega_0) + c\omega_0}{\omega_0\sqrt{k_1 I}} \tag{24}$$

The complete solution is as follows:

$$\varphi = \omega_0 t + \frac{\boldsymbol{T}_{fb} sign(\dot{\varphi} - \omega_0)}{k_1} + \frac{\omega_0}{f_n}e^{-\xi f_n t}[(F\xi + 1)\sin f_n t + (2\xi - F)\cos f_n t] \tag{25}$$

Taking the derivative of the solution with respect to $t$, and omitting the $\zeta^2$ terms:

$$\dot{\varphi} = \omega_0 - \omega_0 e^{-\zeta\omega_n t}[(\xi - F)\sin f_n t + \cos f_n t] \tag{26}$$

According to the above equation, $\dot{\varphi}$ contains a constant component $\omega_0$ and a vibration component $\omega_0 e^{-\zeta\omega_n t}[(\xi - F)\sin f_n t + \cos f_n t]$. If the amplitude of the latter is greater than that of the former, vibration will occur. Therefore, the sufficient condition for the system to exhibit frictional self-excited vibration is that the velocity remains positive, which means:

$$e^{-\zeta\omega_n t}[(\xi - F)\sin f_n t + \cos f_n t] < 1 \tag{27}$$

Considering the critical state, an approximation can be obtained through a Taylor series expansion [22]:

$$F_c \approx \sqrt{4\pi\xi} \qquad (28)$$

Substituting into Equation (24), the critical speed can be calculated as:

$$\omega_c = \frac{\boldsymbol{T}_{fb}sign\left(\dot{\varphi} - \omega_0\right)}{\sqrt{2\pi c \sqrt{k_1 I} - c}} \qquad (29)$$

Based on the preceding analysis, when the driving force exceeds the maximum static friction force, the surfaces begin to slide relative to each other, and the friction force transitions to the relatively lower dynamic friction force. As the deformation of the torsion spring increases, the driving force decreases until the sliding ceases. This process repeats cyclically. The vibration phenomenon induced by this stick–slip process is referred to as self-excited friction vibration. According to Equation (29), when the system's driving speed exceeds this critical value, the overrunning spring clutch will not exhibit self-excited vibrations during operation; hence, self-excited vibration will only occur when $\dot{\varphi} < \omega_0$. In practical applications, the clutch often operates at low rotational speeds, which can lead to self-excited frictional vibrations and the occurrence of unusual noises. In addition, the critical speed $\omega_c$ decreases first and then increases with the increase in damping c; it decreases with the increase in the equivalent stiffness $k_1$.

## 3. Structural Optimization Design

### 3.1. Structural Optimization Analysis

Based on the analysis above, the origin of abnormal noises in an overrunning spring clutch system is attributed to the occurrence of friction-induced self-excited vibrations. In practical engineering applications, it is common to mitigate this issue by using lubricants to transform dry friction into viscous friction, effectively optimizing the system into an overdamped one [23]. Particularly, enhancing the friction between the spring and the central shaft within the clutch system enhances the system's damping coefficient $c$ appropriately and reduces the equivalent stiffness $\mu_0$, thereby lowering the critical speed and preventing the occurrence of unusual noises.

To verify the optimization effects of commonly used methods, three groups of clutch systems with abnormal noise were selected for optimization effect testing experiments. Lubricating grease was added to the three groups of noisy systems A, B, and C, respectively, to ensure sufficient lubrication of the overall system, as shown on the right side of Figure 9. Subsequently, the clutch systems were then installed in the lifting component system, as shown on the left side of Figure 9.

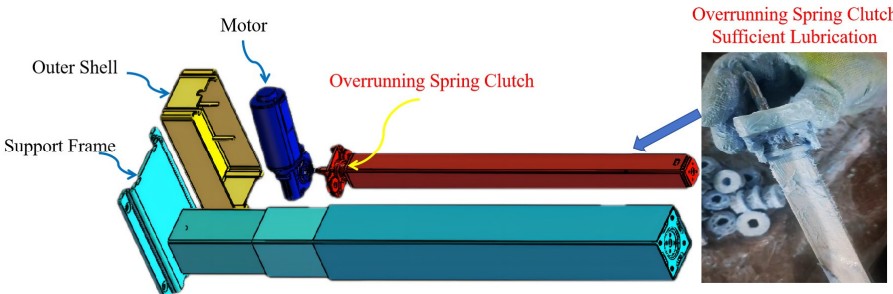

**Figure 9.** Common methods testing experiment.

The experimental temperature for the three experimental systems was 20 °C, and the end of the support bracket remained unloaded. After the lifting tests, the results were obtained and are shown in Table 2, where groups A and C still exhibited abnormal noise, while the abnormal noise phenomenon in group B was resolved, with the proportion of abnormal groups reduced to 66.7% after optimization. These findings suggest that the

lubricant optimization approach has a discernible effect on mitigating abnormal noise in an overrunning spring clutch. By introducing enhanced lubricating oil, dry friction among the clutch components can be converted into viscous friction, thereby augmenting system damping and suppressing the occurrence of abnormal noise.

**Table 2.** Optimization experiment of lubricant for beyond torsion spring clutch system.

|  | **Group A** | **Group B** | **Group C** | **Abnormal Group Proportion** |
|---|---|---|---|---|
| Before lubricant optimization | Unusual noises | Unusual noises | Unusual noises | 3/3 |
| After lubricant optimization | Unusual noises | No noises | Unusual noises | 2/3 |

The optimization method demonstrated only a 33% success rate, indicating certain limitations. Firstly, the continuous variation in the clearance between the spring and the central shaft during the overrunning operation resulted in lubricant loss, resulting in abnormal noise. Secondly, the initial state of the overrunning diaphragm spring clutch involved an interference fit, causing the lubricating oil between the contact surfaces to be easily displaced, thereby leading to dry friction. Therefore, even with a large amount of lubricating oil added during installation, it could only partially suppress abnormal noise.

To address these issues, a textured surface was introduced to the central shaft of the clutch, as depicted in Figure 10a. To determine the shape and width of the texture structure, modeling was performed, as shown in Figure 10b. *T* represents the periodic length dimension of the calculation unit, $h_0$ is the oil film thickness of the friction pair in the lubricated condition, *H* and *W* denote the depth and width dimensions, respectively, while *R* stands for the central shaft's radius of curvature. In order to facilitate one-shot injection molding, the width of the texture structure was selected to be between 0.6 mm and 1.6 mm, the depth of the texture structure was set at 1 mm, and the number of texture structures was chosen to be eight, making it convenient for mold manufacturing and part demolding.

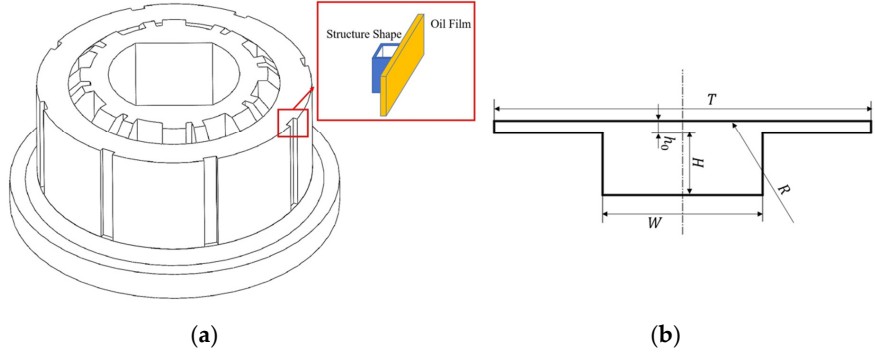

(**a**)  (**b**)

**Figure 10.** Torsional spring structural optimization method and the model of texture structure. Textured surface was introduced to clutch (**a**). The shape and width of texture structure (**b**).

### 3.2. Structure CFD Simulation Analysis

Developing a CFD texture structure simulation model based on the N-S equations using the ANSYS Fluent module, this study employed a controlled-variable approach to investigate the influence of the texture structure width and shape on the fluid domain distribution within the model under fixed rotational speed and identical lubrication conditions. The objective of this analysis was to evaluate the impact of the texture structure width and shape on the surface friction performance.

The analysis employed the Schnerr and Sauer two-phase flow model, focusing on a steady-state, incompressible, and isothermal laminar flow setup. The medium comprised a lubricant and air, with a lubricant density $\rho$ of 960 kg/m$^3$ and a dynamic viscosity $\eta$ of

0.048 kg/(m·s). The upper wall was mobile, the lower wall was stationary. The chosen solution algorithm was Coupled. The initial values for the x- and y-direction velocity and fluid domain pressure were set at 0, and the simulation iterated for 500 cycles.

Figure 11 presents the lubricant distribution for the different lubricant fractions under various widths $W$ of the structure. Different colors denote distinct lubricant volume fractions, with red indicating the highest volume fraction and blue representing the lowest. The black box highlights sections of lubricant film breakage. The lubrication effectiveness was assessed based on the lubricant fraction and the length of the lubricant film breakage. A higher proportion of red indicates enhanced storage capacity for the structure, while a shorter breakage length suggests improved lubrication performance.

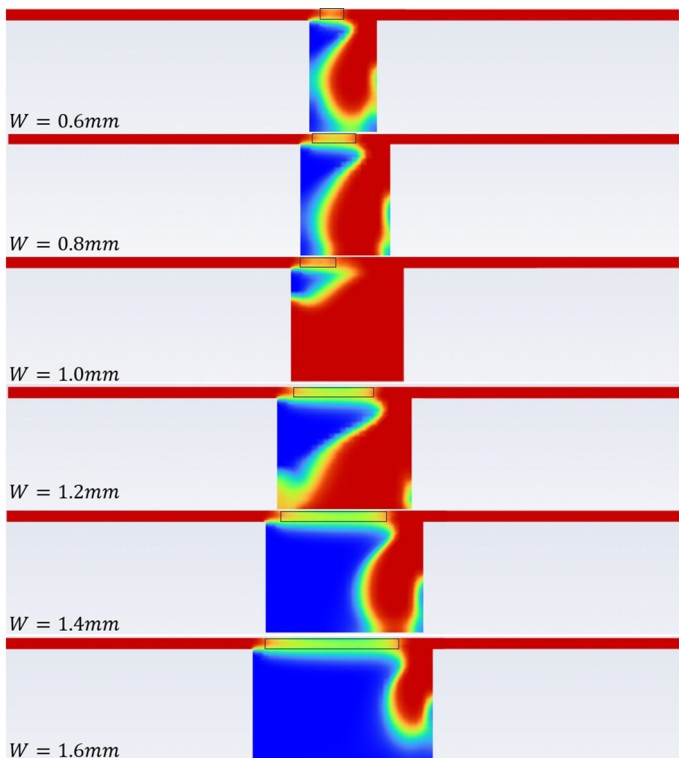

**Figure 11.** Simulation results of different width structures.

Image processing was performed using the Python Pillow library, eliminating the background color and measuring the length of the lubricant film breakage while calculating the red lubricant fraction. The polynomial curve fitting results are shown in Figure 12a,b. As shown in Figure 12a, the length of breakage increased with the widening of the structure width $W$. The increment was gradual when $W$ was less than 1 mm but sharply rose beyond 1 mm. Conversely, as shown in Figure 12b, the lubricant fraction initially increased and then decreased with the increase in the structure width. The fraction continuously grew from a 0.6 mm to 1.0 mm width, reaching its peak around 0.95 mm. Consequently, at $W = 0.95$ mm, the lubricant storage was optimal with a minimal length of lubricant film breakage, indicating the best overall performance.

To mitigate the lubrication's impact on the system's self-locking performance, the structure shape was optimized to a parallelogram, as shown in Figure 13. The lubrication conditions for both working states before and after optimization are solved and compared in Figure 14. It is evident that the lubricant distribution in the optimized model aligned with the different lubrication requirements for the self-locking and surpassing working states, with a longer length of breakage in the self-locking state and shorter ones in the overrunning state.

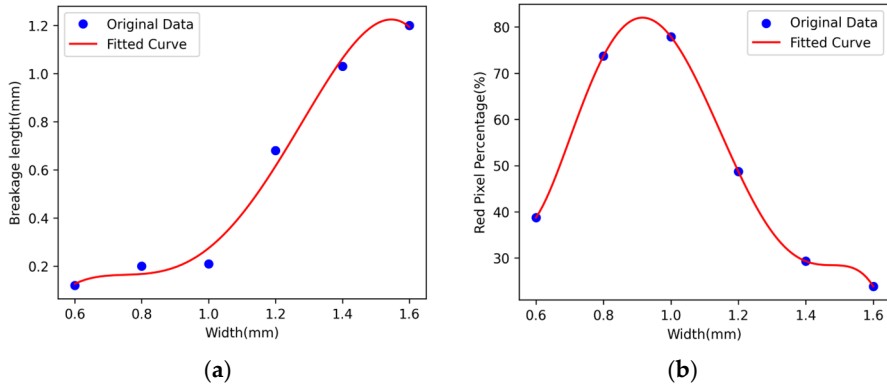

**Figure 12.** Polynomial curve fitting results. The relationship between the length of breakage and the structure width (**a**). The relationship between the red pixel percentage and the structure width (**b**).

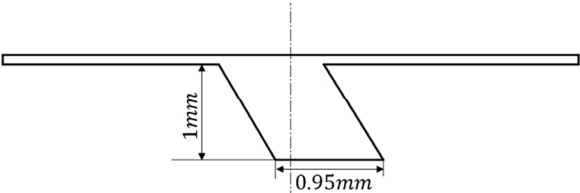

**Figure 13.** Improved texture structure.

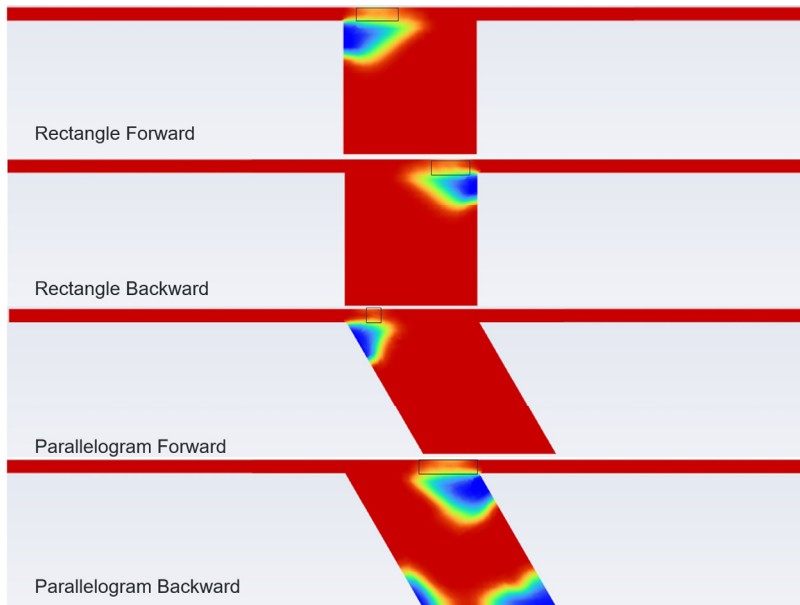

**Figure 14.** Comparison of simulation results before and after improvement.

## 4. Experimental Verification

### 4.1. Experimental Apparatus Design

The experimental setup is shown in Figure 15. The overrunning clutch was fixed to the lower end through a mounting bracket, and the upper end was connected to the leadscrew. The leadscrew was driven by a stepper motor. In front of the clutch assembly, there were high-speed cameras and a sound sensor. The camera could capture 300 frames per second, and the sound sensor had a sampling frequency of 48 kHz. The high-speed camera, sound sensor, and stepper motor were all connected to a host computer, allowing real-time control and data acquisition of both sound and image information.

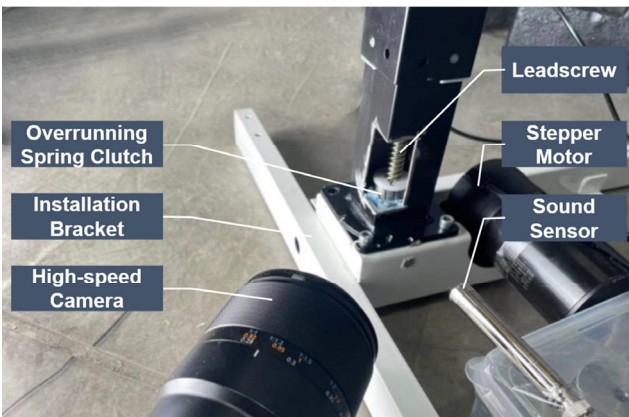

**Figure 15.** Schematic diagram of the experimental platform structure.

*4.2. Experimental Analysis*

Figure 16 shows a series of consecutive frames of the improved overrunning spring clutch captured by the high-speed camera at a 160-times-reduced speed. In comparison to Figure 3b, the improved clutch did not exhibit obvious ghosting.

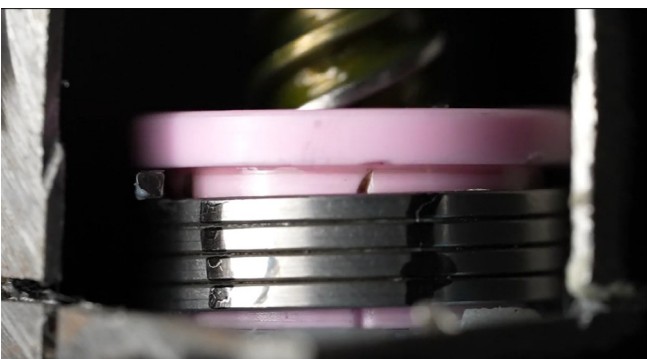

**Figure 16.** High-speed camera images captured of the improved overrunning spring clutch.

To further validate the improvement, tests were conducted on three sets of overrunning spring clutches, both before and after the enhancements. Using the method outlined in Section 2.1, the blur calculation was performed on the three sets of videos before and after the improvements, yielding the results shown in Figure 17a–f. Figure 17a,c,e represent the blur levels before the improvements, while Figure 17b,d,f represent the blur levels after the enhancements.

As shown in Figure 17a, the blur level exhibited periodic fluctuations at approximately $1.80 \times 10^{10}$. As shown in Figure 17c, the blur level had periodic fluctuations at around $1.88 \times 10^{10}$, and as shown in Figure 17e, the blur level showed periodic fluctuations at about $2.00 \times 10^{10}$. On the other hand, as shown in Figure 17b, the blur level remained stable at approximately $1.69 \times 10^{10}$, as shown in Figure 17d, it remained stable at around $1.75 \times 10^{10}$, and as shown in Figure 17f, the blur level remained stable at about $1.68 \times 10^{10}$. It is evident that the image motion blur values significantly decreased after the improvements, and the blur curves no longer exhibited periodic fluctuations. This indicates a noticeable reduction in the vibration of the spring after the enhancements.

Figure 18a–f depicts the audio waveform graphs corresponding to three sets of videos before and after the improvement of the clutch, obtained by conducting a fast Fourier transform. Figure 18a,c,e represent the audio waveforms before the improvement, while Figure 18b,d,f represent the audio waveforms after the improvement. The average amplitude index of the audio waveforms before the improvement was 452.0 Pa, and the waveforms exhibited significant periodic fluctuations. After the improvement, the average amplitude index of the audio waveforms was 179.7 Pa, which was reduced by 60.2% com-

pared to before, and the waveforms no longer exhibited significant periodic fluctuations. Therefore, the improvement significantly suppressed the abnormal noise in the overrunning spring clutch. Based on the analysis from Figures 16–18, the spring modification method proposed in this paper effectively reduces spring-induced frictional self-excited vibrations, thereby preventing the occurrence of abnormal noise in the overrunning clutch during overrunning operation.

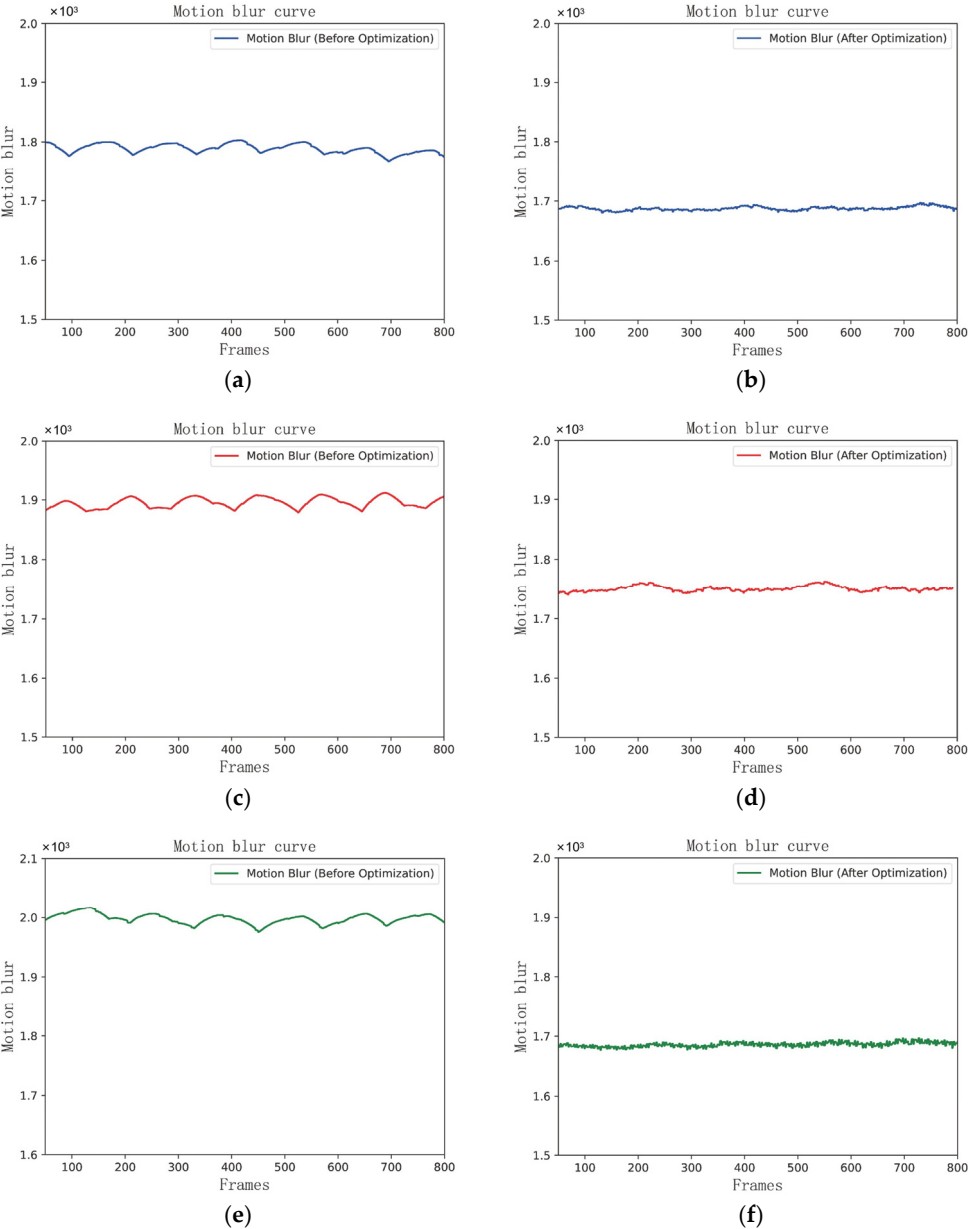

**Figure 17.** Comparison of blur curves before and after clutch improvement. The motion blur of the first group of overrunning spring clutches before optimization (**a**). The motion blur of the first group of overrunning spring clutches after optimization (**b**). The motion blur of the second group of overrunning spring clutches before optimization (**c**). The motion blur of the second group of overrunning spring clutches after optimization (**d**). The motion blur of the third group of overrunning spring clutches before optimization (**e**). The motion blur of the third group of overrunning spring clutches after optimization (**f**).

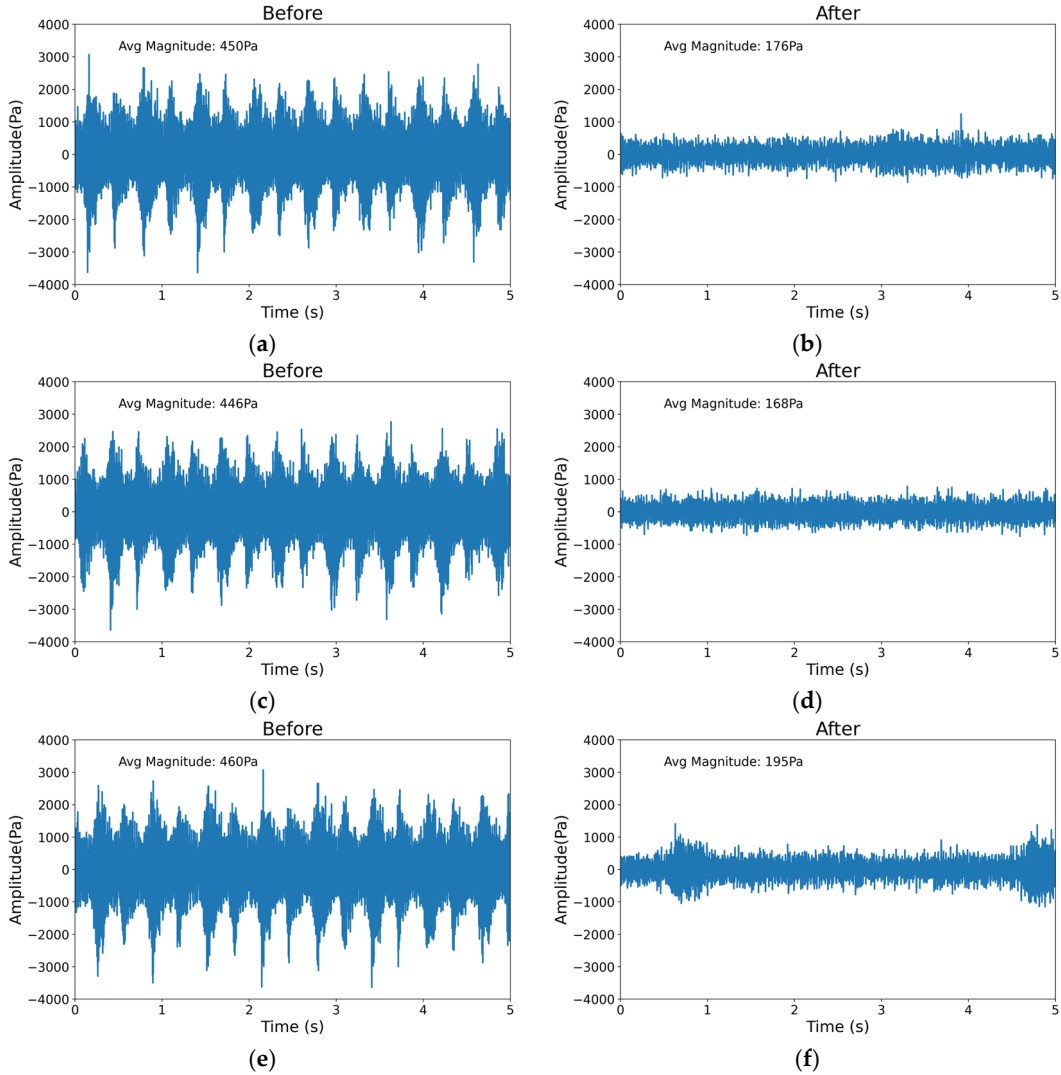

**Figure 18.** Comparison of working noise curve before and after improvement. The audio waveform of the first group of overrunning spring clutches before optimization (**a**). The audio waveform of the first group of overrunning spring clutches after optimization (**b**). The audio waveform of the second group of overrunning spring clutches before optimization (**c**). The audio waveform of the second group of overrunning spring clutches after optimization (**d**). The audio waveform of the third group of overrunning spring clutches before optimization (**e**). The audio waveform of the third group of overrunning spring clutches after optimization (**f**).

## 5. Conclusions

To suppress the phenomenon of abnormal noise in overrunning spring clutches, this study conducted a dynamic analysis of these clutches and concluded that the mechanism behind the abnormal noise is the system's self-excited frictional vibration. Further optimization of the system's damping using surface texturing was demonstrated through experiments to effectively suppress the generation of abnormal noise. After the improvement, the average amplitude index of the audio waveforms was reduced by 77.32% compared to before. As most spring clutch seats are manufactured using injection molding, the currently widely adopted laser processing for micron-level surface texturing is inefficient. However, this study proposes a texturing structure with a width of 0.95 mm and a depth of 1 mm, which is suitable for one-shot injection molding and provides good noise suppression, meeting the practical engineering application requirements. The theoretical research and experimental analysis in this paper led to the following conclusions:

(1)    The abnormal noise during the reverse overrunning process of overrunning spring clutches originates from the self-excited frictional vibration caused by the stick–slip phenomenon between the spring and the clutch seat.

(2)    Based on the system's dynamic model of overrunning spring clutches, the generation of self-excited frictional vibration in the clutch system is related to the rotational speed. The critical speed is directly proportional to the system's damping, and changing the system's damping can alter the critical speed at which frictional self-excited vibration occurs.

(3)    Optimized surface texturing can increase system damping and effectively suppress abnormal noise while reducing its impact on clutch performance.

**Author Contributions:** Conceptualization, J.Z. (Jie Zhou) and Z.Q.; methodology, Z.Q.; validation, Z.Q. and J.Z. (Jie Zhou); formal analysis, Z.Q.; investigation, H.Z.; resources, J.Z. (Jianming Zhan); data curation, J.Z. (Jie Zhou); writing—original draft preparation, Z.Q.; writing—review and editing, J.Z. (Jie Zhou); visualization, H.Z.; project administration, J.Z. (Jianming Zhan); funding acquisition, J.Z. (Jianming Zhan). All authors have read and agreed to the published version of the manuscript.

**Funding:** This work was supported by the following projects: Research on Low-Temperature Adaptability Technology of Large Cold Storage Warehouse Robots (grant No. 2023S072), the Key Technology Research of Intelligent Stealth Forklift Mobile Robots (grant No. 2023Z041), and the Ningbo Natural Science Foundation (grant No. 2021J165).

**Data Availability Statement:** Data are unavailable due to privacy or ethical restrictions.

**Acknowledgments:** The authors would like to thank Zejiong Wei, Zhechen Yang, and Zhaotian ding for their assistance with the experiments.

**Conflicts of Interest:** The authors declare no conflicts of interest.

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
