# Peer review of "Study on Dynamics of Overrunning Spring Clutches and Suppression Methods for Their Abnormal Noise"

_actuators, doi:10.3390/act13050165_

Round 1

Reviewer 1 Report

Comments and Suggestions for Authors

This paper analysis the noise in clutches due to friction which causes stick-slip and therefore noise. The authors propose to solve this problem by adding damping by the appropriate surface texture. The experimental results confirm the effectiveness of their approach.

Feedback points:

·         The noise is produced by so-called stick-slip of the friction force. Please add this in the description and in the summary of the conclusions (the authors call this “self-excited frictional vibration”).

·         The equation of motion (18) includes the constant value of T_fb, which is a constant number according to (16). This friction force mainly depends on the friction coefficient and the normal force. If this reviewer understands this equation correctly then this friction force is constant. So, the question arises how the equation of motion (18) can describe the rotation which is not constant because of the stick-slip effects of the friction which is the source of the noise (the authors call this “self-excited frictional vibration”). Author should explain which term describes the stick-slip effect. Also, if T_fb is a dissipative friction force then one would expect that the sign of the rotational speed must be included in equation (16) or at least in (18).

Reviewer 2 Report

Comments and Suggestions for Authors

1\About 50% of the references in the article are 10-20 years old. This raises questions about the relevance of the research. Therefore, it is necessary to modify the references to highlight the research relevance.

2\ Please explain the novelty of the proposed solution compared to existing solutions. There is no obvious improvement in the existing explanation, and the analysis process lacks depth, which is not sufficient to fully demonstrate the innovation of the design scheme in this article. It is recommended to increase the effectiveness comparison with existing research.

3\The improved texture structure of ‘1mm × 0.95 mm’, please provide the specific optimization process and optimization indicators for these two parameters?  Optimization indicators? Optimization flowchart?

4\ The quality of the images in the text needs further optimization.

5\ References 1-4 are not indicated.

6\There are many formatting issues in the manuscript. 

In line 236 ‘equations (10) and (8)’ should be changed as ‘Equations (10) and (8)’; And the same problem also exists in other places;

In line 283 ‘Where’ should be changed as ‘where’, and this line does not require indentation;

Comments on the Quality of English Language

/

Round 2

Reviewer 2 Report

Comments and Suggestions for Authors

1\ In line 93, the author point out that “laser processes are less efficient than mechanical methods [18] and are not suitable for large-scale engineering applications”. Please further explain the applicability of this discussion.

2\ In line 174, Figure 2.5 ?

3\Please explain how Figure 4 is obtained. Please explain (a) and (b) of Figure 4 separately.

4\How to compare Figure 4 and Figure 2 to obtain this conclusion “the vibration of the torsion spring highly correlates with the timing of the abnormal noise”.

5\ The subgraphs in the figure should be explained in the title, such as (a) and (b) of Figure 12; (a)-(f) of Figure 17.

Comments on the Quality of English Language

Line199-line202, Line378-Line381, there are grammar problems with the paragraphs.  There are also some grammar and formatting issues in other places
